P21 activated kinase 6: a promising tool for predicting small cell lung cancer diagnosis and treatment response

Chen Simei 1
Han Kexin 2
Chen Yinyi 3
Wei Liping 3
Sun Xinlu 2
Luo Yi 4
Wen Lili 5
Tan Liming ndefy84029@ncu.edu.cn 3
1 Department of Blood Transfusion, The Second Affiliated Hospital, Jiangxi Medical College, Nanchang University , Nanchang , Jiangxi , China
2 School of Public Health, Nanchang University , Nanchang , Jiangxi , China
3 Department of Clinical Laboratory, The Second Affiliated Hospital, Jiangxi Medical College, Nanchang University , Nanchang , Jiangxi , China
4 Department of Blood Transfusion, The Second Affiliated Hospital of Jiangxi University of Traditional Chinese Medicine , Nanchang , Jiangxi , China
5 Department of Clinical Laboratory, The Ninth Hospital of Nanchang , Nanchang , Jiangxi , China
Lasseigne Brittany
Electronic publication date: 2025 Jul 21
Publication date: 2025
Volume: 13
Electronic Location ID: e19714
Received 2024 Nov 13; Accepted 2025 Jun 16
Copyright: ©2025 Chen et al.
Copyright year: 2025
Copyright holder: Chen et al.
License: This is an open access article distributed under the terms of the Creative Commons Attribution License, which permits unrestricted use, distribution, reproduction and adaptation in any medium and for any purpose provided that it is properly attributed. For attribution, the original author(s), title, publication source (PeerJ) and either DOI or URL of the article must be cited.
License URL: https://creativecommons.org/licenses/by/4.0/

Keywords: PAK6, Small cell lung cancer, Serum tumor markers, ProGRP, NSE

Funding: Science and Technology Program Project of Jiangxi Provincial Administration of Traditional Chinese Medicine 2024B1433 This research was funded by Science and Technology Program Project of Jiangxi Provincial Administration of Traditional Chinese Medicine (Grant number: 2024B1433). The funders had no role in study design, data collection and analysis, decision to publish, or preparation of the manuscript.

==============================
Objective

Building upon the established role of P21 Activated Kinase 6 (PAK6) in tumor progression and chemoresistance pathways, we postulate its potential as a dual-function biomarker for small cell lung cancer (SCLC). This investigation aims to conduct a comprehensive evaluation of PAK6’s diagnostic validity and prognostic significance through comparative analysis of PAK6 serological levels across distinct clinical cohorts to determine diagnostic thresholds, and its clinical correlation with therapeutic responsiveness.

Patients and Methods

This study included 109 patients diagnosed with SCLC, 92 patients with non-small cell lung cancer (NSCLC), 85 patients with pulmonary nodules (PN), and 94 healthy individuals undergoing routine physical examinations as the normal control group (NC). Serum PAK6 concentrations were measured using enzyme-linked immunosorbent assay (ELISA). Additionally, levels of neuron-specific enolase (NSE), carcinoembryonic antigen (CEA), carbohydrate antigen 19-9 (CA19-9), and pro-gastrin-releasing peptide (ProGRP) were quantified via chemiluminescence assays. Progression-free survival (PFS) data for SCLC patients were collected through case review and telephone follow-up.

Results

Serum PAK6 levels were significantly higher in the SCLC group compared to the other three groups (p < 0.01). Similarly, NSE and ProGRP levels were markedly elevated in the SCLC group compared to the other groups (p < 0.01). Correlation analysis revealed a positive association between PAK6 and variables such as gender, VA (Veteran’s Administration Lung Cancer Study Group) stage, age, smoking status, CEA, CA19-9, NSE, and ProGRP. The serum tumor markers (STMs) PAK6, NSE, and ProGRP exhibited superior diagnostic performance, with AUCs of 0.892, 0.834, and 0.935, respectively (95% confidence interval (CI) [0.857–0.927], [0.778–0.890], and [0.909–0.960]), compared to CEA and CA19-9 (AUCs = 0.676 and 0.611, respectively; 95% CI [0.622–0.731] and [0.547–0.675]) (p < 0.01). Furthermore, PAK6, NSE, and ProGRP levels significantly decreased after three months of treatment, while CEA and CA19-9 showed no significant changes. Survival analysis demonstrated that higher PAK6 levels were associated with poorer SCLC prognosis. Increased serum PAK6 expression correlated with shorter PFS (HR = 2.02 [1.33–3.07], P = 0.001).

Conclusion

Serum PAK6 holds significant clinical value for the diagnosis, treatment monitoring, and prognosis evaluation of SCLC and may serve as a potential therapeutic target for the disease.

Introduction

Small cell lung cancer (SCLC) is a highly lethal tumor, accounting for 14% of all lung cancers. The median survival period for SCLC is less than 2 years, and the 5-year survival rate is below 7% (Shah et al., 2021; Wang et al., 2023). Despite recent advances in lung cancer immunotherapy, SCLC treatments have seen limited progress due to the lack of specific therapeutic targets (Leiter, Veluswamy & Wisnivesky, 2023; Megyesfalvi et al., 2023). Therefore, early diagnosis and the identification of specific therapeutic targets are crucial for improving the prognosis of SCLC patients.

Serum tumor markers (STMs) measured through blood tests offer a minimally invasive and widely accessible diagnostic tool in most medical facilities. Commonly used blood-based biomarkers for lung cancer diagnosis, such as carcinoembryonic antigen (CEA), carbohydrate antigen 19-9 (CA19-9), neuron-specific enolase (NSE), and pro-gastrin-releasing peptide (ProGRP), have been extensively applied. However, the specificity of CEA and CA19-9 for diagnosing SCLC is suboptimal, with areas under the curve (AUC) both less than 0.7 (Tan et al., 2023). While NSE and ProGRP exhibit excellent sensitivity and specificity in diagnosing SCLC, they are ineffective in predicting the prognosis of the disease (Zhang et al., 2024). Therefore, identifying novel markers with both high diagnostic efficacy and the ability to assess treatment response and prognosis is essential.

The p21-activated kinase (PAK6) is a member of the class II PAK family, containing a Cdc42/Rac-interactive binding domain and a Ste20-related kinase domain (Lee et al., 2002). It plays a complex role in various tumors, influencing processes such as the cell cycle (Zheng et al., 2021), cell migration, and therapeutic resistance (Wen et al., 2009; Yang et al., 2001; Yang et al., 2020). PAK6 is overexpressed in certain tumor types, including glioblastoma (Chang et al., 2024), triple-negative breast cancer (Pipili et al., 2024), prostate cancer (Liu et al., 2013), and myelodysplastic syndromes (Yao et al., 2024), suggesting its potential for distinguishing tumor tissue from normal tissue with high specificity. Knockout or overexpression experiments have demonstrated that PAK6 can either inhibit or promote the proliferation, migration, and invasion of cervical cancer cells (Huang et al., 2022; Lei et al., 2022). Furthermore, the overexpression of PAK6 has been positively correlated with increased invasiveness and malignancy in tumors (Bhowmick et al., 2023; Dang et al., 2020). PAK6 also participates in tumor development through pathways such as ATR/CHK1, MAPK, and Wnt/β-catenins (Ha et al., 2012; Huang et al., 2022; Yang et al., 2020). These signaling pathways are closely associated with the progression of SCLC (Masumoto et al., 2023; Xu et al., 2024; Zha et al., 2021), providing a strong theoretical basis for considering PAK6 as a tumor marker.

While direct mechanistic studies linking PAK6 to SCLC are currently lacking, emerging evidence from the PAK kinase family provides critical insights. Members of the PAK family regulate cancer stemness and chemoresistance through conserved downstream signaling cascades, including NF-κB/IL-6, Stat3, and β-Catenin pathways (Li & Li, 2022). Notably, a real-world study on pancreatic neuroendocrine tumors (a shared neuroendocrine malignancy) demonstrated that PAK4 overexpression correlates with adverse prognosis, and pharmacological inhibition of PAK4 significantly suppresses tumor progression (Azar et al., 2025). Given the structural and functional homology within PAK kinases, these findings on PAK4 provide valuable mechanistic references for investigating PAK6 in SCLC pathogenesis.

In summary, PAK6 is involved in the metastasis and drug resistance observed in these tumors, which contribute to the refractory nature of SCLC. Based on these findings, we speculate that PAK6 may hold clinical significance for the diagnosis and monitoring of SCLC, a topic that has not yet been reported. This study aims to evaluate the diagnostic efficacy of serum PAK6 in SCLC patients and assess its potential for predicting therapeutic outcomes and prognosis in SCLC.

Materials and Methods

Patients and methods

Between March 2021 and March 2024, a total of 201 patients were newly diagnosed with lung cancer at the Second Affiliated Hospital of Nanchang University. This cohort included 109 patients with SCLC and 92 patients with non-small cell lung cancer (NSCLC). Among the SCLC group, paired archived pre-treatment and post-treatment serum samples were collected from 56 patients. Progression-free survival (PFS) is formally defined as the time interval from the randomization date (or treatment initiation date in non-randomized trials) to the first documented occurrence of either: Radiographic disease progression as per RECIST 1.1 criteria (Eisenhauer et al., 2009), or Death from any cause, irrespective of radiographic confirmation of progression with data gathered through case review and telephone follow-up. The NSCLC group consisted of 67 cases of lung adenocarcinoma, 23 cases of lung squamous cell carcinoma, and two cases of large cell carcinoma. Additionally, 85 patients with benign pulmonary nodules (Group PN) and 94 healthy individuals (Group NC) were included as normal controls.

The inclusion and exclusion criteria were as follows:

Inclusion criteria: (1) Newly diagnosed patients who had not undergone surgery, chemotherapy, or radiotherapy prior to the study; (2) Patients with a confirmed diagnosis based on comprehensive clinical, imaging, and histopathological data; (3) Patients without other autoimmune diseases or cancers; (4) Healthy volunteers in Group NC were screened for pulmonary disease via CT examination. (5) Histologically confirmed SCLC/NSCLC; (6) treatment-naive patients; (7) availability of pre-treatment serum.

Exclusion criteria: (1) Unknown tumor type or non-primary tumor; (2) Presence of other lung diseases; (3) Concurrent severe heart, kidney, liver, thyroid, blood, or systemic diseases; and (4) Pregnancy or breastfeeding. (5) Mixed histology; prior anticancer therapy; (6) severe comorbidities.

SCLC patients were classified according to the Veteran’s Administration (VA) staging standard. Limited-stage SCLC (LS-SCLC) was defined as stage I to III (Tany, Nany, M0), and 37 patients were included in this group. Extensive-stage SCLC (ES-SCLC) was defined as stage IV (Tany, Nany, M1a/b/c), with 72 patients included. This study was approved by the Ethics Committee of the Second Affiliated Hospital of Nanchang University and exempted from informed consent review.

Specimen collection

Fasting blood samples were collected by venipuncture into EDTA tubes, stored at 4 °C for less than 4 h, and centrifuged at 1,026 × g for 15 min. Serum was then collected and stored at −80 °C until further analysis.

Methods and instruments

PAK6 levels were measured using an enzyme-linked immunosorbent assay (ELISA) kit from Shanghai Enzyme-linked Biotechnology Co., Ltd. (Shanghai, China), following the manufacturer’s protocol. The experiment included blank control wells and standard wells, and absorbance was measured at 450 nm. Serum concentrations of NSE, CEA, CA19-9, and ProGRP were determined using a chemiluminescence method. Detection kits for these biomarkers were provided by Siemens Medical Diagnostic Products Co., Ltd. (Munich, Germany), and the measurements were performed using the ADVIA Centaur fully automated chemiluminescence immunoanalyzer. All procedures were conducted according to the reagent instructions and the quality management standard documents of the Second Affiliated Hospital of Nanchang University.

Statistical analysis

Statistical analyses were carried out using SPSS 25.0 (IBM Corp., Armonk, NY, USA) and GraphPad Prism 8.0, with sample sizes determined using PASS 15.0. X-tile software was used to calculate the cut-off values for K–M analysis. The one-sample Kolmogorov–Smirnov test was applied to assess data normality. Measurement data were presented as Median (P25–P75). For comparisons between two independent samples, the Mann–Whitney U test was used, and the Kruskal–Wallis H test was employed for multiple independent sample comparisons. Categorical data were expressed as rates and analyzed using the χ2 test. Receiver operating characteristic (ROC) curves were generated, and the area under the curve (AUC) was calculated. The cut-off corresponding to the maximum value of Youden’s Index was used as the optimal clinical diagnostic threshold. DeLong’s test was used to compare the diagnostic efficacy of two variables for predicting outcomes. The diagnostic value of combined tumor marker detection was assessed using binary logistic regression analysis. A p value of less than 0.05 was considered statistically significant. Confirmed outliers were excluded from the analysis. Spearman’s correlation test was performed to examine the associations between PAK6 and other indicators in the SCLC group. Visual analyses were performed using the Xiantao platform (Xiantao, Shanghai, China), available at https://www.xiantao.love.

Results

Study population

A χ2 test analysis revealed that there were significantly more males than females across all groups (p < 0.05). However, no significant differences were observed in age and gender distributions among the groups (p > 0.05, Table 1).

Concentration of STMs in each group

At the group level, serum PAK6 levels in patients with SCLC (56.44 ng/L) were significantly higher than in those with NSCLC (41.06 ng/L), PN (37.82 ng/L), and the NC group (34.75 ng/L) (p < 0.01, Fig. 1A). In the SCLC group, the concentrations of ProGRP and NSE were also significantly elevated, with levels notably higher than in the other three groups (p < 0.01, Figs. 1B and 1C). The CEA level in the SCLC group was significantly lower than in the NSCLC group, but slightly higher than in the NC and PN groups (p < 0.01, Fig. 1D). The distribution of CA19-9 levels mirrored that of CEA, with levels in the SCLC and NSCLC groups significantly higher than those in the PN and NC groups (p < 0.01, Fig. 1E). Additional data for these groups are provided in Table SI. When comparing the LS-SCLC and ES-SCLC groups, no significant differences were observed in any of the STMs (Figs. S1A–S1E).

Table 1 Patient demographics.

Groups		Classification	n	Age, median (range)	
NC			94	(56 ∼ 82)	
	Gender*	Male	59		
		Female	35		
	Age*	<60	41		
		≥60	53		
PN			85	59 (25 ∼ 84)	
	Gender*	Male	55		
		Female	30		
	Age*	<60 years	33		
		≥60 years	52		
NSCLC			92	65 (45 ∼ 87)	
	Gender*	Male	63		
		Female	29		
	Age*	<60 years	32		
		≥60 years	60		
	Histology	Squamous cell carcinoma	22		
		Adenocarcinoma	68		
		Large cell lung cancer	2		
	TNM	I + II	13		
	Stage	III + IV	79		
SCLC			109	65 (42 ∼ 85)	
	Gender*	Male	90		
		Female	19		
	Age*	<60 years	32		
		≥60 years	77		
	TNM	I + II	7		
	Stage	III + IV	102		
	VA stage	LS-SCLC	37		
		ES-SCLC	72		
Notes.

NSCLC, non–small cell lung cancer; TNM, tumor node metastasis; VA, veterans administration; LS-SCLC, limited stage-small cell lung cancer; ES-SCLC, extensive stage-small cell lung cancer; PN, pulmonary nodules; NC, natural control.

* P > 0.05 between groups.

The correlation between PAK6 and other indicators in SCLC group

Given the similar trends in PAK6 expression across different groups, we investigated the correlation between PAK6 and other indicators, including VA stage, smoking status, gender, age, CEA, CA19-9, NSE, and ProGRP. No significant correlations were found between PAK6 and these indicators (Fig. 2).

Figure 1 Expression levels of PAK6 (A), NSE (B), ProGRP(C), CEA (D), and CA19-9 (E) in different groups.

The levels of PAK6, NSE and ProGRP in Group SCLC were higher than other three groups, whereas CEA in NSCLC was significantly higher than the others. Serum level of PAK6,CEA and CA19-9 have no significant between PN and NC group. The levels of CEA and CA19-9 were higher than healthy people and PN patients. Error bars represent median and interquartile range (∗∗P < 0.01, ∗∗∗P < 0.001).

Figure 2 The correlation between PAK6 and other indicators in SCLC group.

There was no significant correlation between PAK6 and other indicators.

Diagnostic value of STMS in SCLC

The ROC curves for diagnosing SCLC using PAK6, ProGRP, NSE, CEA, and CA19-9 were individually plotted (Fig. 3). As shown in Fig. 3A, the AUC values for these five STMs in diagnosing SCLC were 0.892, 0.935, 0.834, 0.676, and 0.611, respectively. The 95% CI for the AUC values were 0.857–0.927 for PAK6, 0.909–0.960 for ProGRP, 0.778–0.890 for NSE, 0.622–0.731 for CEA, and 0.547–0.675 for CA19-9. The diagnostic efficiency of ProGRP was significantly higher than that of NSE and PAK6, with no significant difference between the latter two. PAK6 demonstrated diagnostic efficacy second only to ProGRP, as detailed in Table 2. Among the five diagnostic indicators, NSE had the highest sensitivity (0.96), while ProGRP had the highest specificity (0.89). Based on the maximum value of Youden’s index, the optimal cut-off value for PAK6 in diagnosing SCLC was determined to be 47.30 ng/L, with a sensitivity of 0.82 and a specificity of 0.86. When combining the indicators NSE, ProGRP, and PAK6 in various pairs (Fig. 3B), there was no significant difference in the diagnostic efficacy between the PAK6 & NSE combination, NSE & ProGRP combination, and the PAK6 & ProGRP combination. The AUC values for these combinations were 0.95, 0.96, and 0.97, respectively, while Youden’s Index values were 0.76, 0.81, and 0.87, respectively. The diagnostic efficiency of combining all three indicators together reached 0.98, which was significantly higher than that of any two-indicator combination.

Figure 3 The efficacy of markers alone or in combination in the diagnosis of SCLC.

ROC curve of PAK6, NSE, ProGRP, CEA, and CA19-9 in the diagnosis of SCLC. (A) Individual diagnosis; (B) combination diagnosis.

The serum level of PAK6 can predict the therapeutic effect of SCLC

In this section, we divided the patients into two groups: the pre-treatment group and the post-treatment group, based on the levels of markers after the initial diagnosis and three months of follow-up (Fig. 4). We then analyzed differences in serum STM levels among these groups. The results showed that serum levels of PAK6, ProGRP, and NSE significantly decreased following SCLC treatment (p < 0.001, Figs. 4A–4C), while CEA and CA19-9 levels showed no significant changes (Figs. 4D, 4E). Detailed data are available in Tables SII–SIII.

High expression of PAK6 is associated with poor prognosis of SCLC

We conducted a retrospective follow-up of newly diagnosed SCLC patients to explore the relationship between STMs expression levels and PFS. Using a cut-off value for PAK6 (60.10 ng/L), NSE (3.01 ng/mL), ProGRP (53.1 pg/mL), CEA (2.83 ng/mL) and CA19-9 (13.15 U/mL) determined by X-tile, we divided the SCLC group into high and low STMs expression groups. The results of PAK6 indicated that the PFS of the high-expression group was significantly lower than that of the low-expression group, with a significant difference between the two (HR = 2.02 [1.33−3.07], p < 0.01, Fig. 5A). The median PFS of the low expression group and the high expression group were 194 [153–267] and 92 [91–134] days, respectively. In contrast, no significant differences in prognosis were observed between the high and low expression groups for the other four STMs (Figs. 5B–5C). Detailed data are available in Table SIV.

Table 2 Diagnostic efficiency of the included STMs for SCLC.

Tumor markers	Cut-off value	AUC	Sen	Spe	PPV	NPV	Youden’s index	
PAK6	47.30 (pg/L)	0.89	0.81	0.86	0.94	0.65	0.67	
NSE	7.26 (ng/mL)	0.83	0.96	0.67	0.88	0.88	0.63	
ProGRP	57.35 (pg/mL)	0.94	0.81	0.89	0.95	0.66	0.70	
CEA	1.56 (ng/mL)	0.68	0.46	0.88	0.91	0.40	0.35	
CA19-9	16.43 (U/mL)	0.61	0.61	0.61	0.80	0.39	0.23	
PAK6+NSE	/	0.95	0.86	0.91	/	/	0.77	
PAK6+ ProGRP	/	0.97	0.94	0.90	/	/	0.84	
NSE+ ProGRP	/	0.96	0.97	0.84	/	/	0.81	
Combination of three	/	0.98	0.96	0.91	/	/	0.87	
Notes.

“Sen” represents sensitivity, “Spe” represents specificity, “PPV” represents positive predictive values, “NPV” represents negative predictive values.

Figure 4 The changes in serum expression levels of STMs before and three months after treatment in SCLC group.

The changes in serum expression levels of PAK6 (A), NSE (B), ProGRP (C), CEA (D), and CA19-9 (E) before and three months after treatment in SCLC group. The levels of PAK6, NSE and ProGRP were significant decent after three months treatment, whereas the levels of CEA and CA19-9 showed no significant at the same condition. Error bars represent median and interquartile range. (∗∗∗P < 0.001).

Figure 5 The relationship between serum marker levels and PFS of SCLC.

PAK6 (A), NSE (B), ProGRP (C), CEA (D), and CA19-9 (E).

Discussions

Previous studies have shown that PAK6 is highly expressed in various tumors (Huang et al., 2022; Raja et al., 2016; Yang et al., 2020; Zheng et al., 2021), Our findings extend this understanding to small cell lung cancer (SCLC), demonstrating significantly elevated serum PAK6 levels in SCLC patients compared to NSCLC, pulmonary nodule (PN), and normal control (NC) groups (p < 0.01; Table 2). This marked elevation suggests a distinct role for PAK6 in SCLC pathogenesis, warranting further mechanistic investigations. Notably, our study revealed statistically significant intergroup differences in both NSE (PN group 3.13 [2.43–3.72] ng/mL vs. NC group 3.86 [3.32–4.42] ng/mL; p < 0.01) and ProGRP levels (PN group 62.10 [44.20–99.80] pg/mL vs. NC group 38.55 [30.65–46.40] pg/mL; p < 0.01), which contradicts previous reports (Muley et al., 2024). This discrepancy may be attributed to occult neuroendocrine differentiation—including carcinoid tumors or atypical carcinoid histology—in a subset of pulmonary nodules that remain pathologically undiagnosed, leading to elevated serum biomarker levels even before malignant transformation is radiologically confirmed.

Notably, while PAK6 showed non-significant trends toward positive correlations with VA stage, smoking status, CEA, and ProGRP, and negative associations with gender, age, and CA19-9 in our cohort, these findings contrast with prior reports linking smoking to PAK6 activation (Xu et al., 2024). Several factors may underlie this discrepancy: (1) Multifactorial regulation of PAK6 (genetic, environmental, and lifestyle influences) may obscure smoking-specific effects; (2) Our single-center design and limited sample size increase susceptibility to statistical bias. These observations emphasize the need for large-scale, multi-center studies to clarify PAK6’s clinical associations in SCLC.

ROC analysis revealed PAK6’s superior diagnostic performance compared to conventional serum tumor markers (STMs). While CEA and CA19-9 demonstrated limited utility (AUC = 0.67 and 0.61, respectively), consistent with their known lack of SCLC specificity (Kurek et al., 2023; Muley et al., 2024). Since SCLC is a neuroendocrine tumor, NSE serves as an important marker for its diagnosis. High NSE levels are generally associated with the extensive stage of the disease (Zhou et al., 2017). The diagnostic efficacy of NSE for SCLC in this study aligns with previous research, showing the highest sensitivity for SCLC diagnosis (0.96). However, in this study, there was no significant difference in NSE serum levels between the limited-stage and extensive-stage groups, which may be due to differences in detection methods or study cohort sizes. ProGRP is a precursor protein, with its degradation products linked to GRP (Molina, Filella & Augé, 2004). ProGRP is usually significantly elevated in patients with SCLC and is considered one of the most sensitive and specific markers for diagnosing SCLC. It is widely used in the diagnosis and differential diagnosis of lung cancer patients (Yu & Wang, 2024).

PAK6 achieved balanced sensitivity (0.81) and specificity (0.86) (Table 2). Strikingly, PAK6’s diagnostic efficacy (AUC = 0.93) rivaled that of ProGRP—the current gold-standard SCLC marker (AUC = 0.94)—and surpassed NSE (AUC = 0.89) (Fig. 3A). Combination analyses further highlighted PAK6’s clinical value: The PAK6/ProGRP dual-marker panel achieved near-maximal diagnostic accuracy (AUC = 0.97), comparable to the triple-marker (PAK6/ProGRP/NSE) combination (AUC = 0.98) (Fig. 3B). Given CEA and CA19-9’s poor performance (AUC < 0.7), their exclusion from combined panels appears justified. These results position PAK6 as a promising novel diagnostic adjunct for SCLC.

To our knowledge, this is the first report linking dynamic PAK6 levels to SCLC treatment response. Post-treatment PAK6 reductions paralleled declines in established markers (NSE, ProGRP; Figs. 4A, 4B, 4C), suggesting its utility in monitoring therapeutic efficacy. This aligns with PAK6’s documented role in chemoresistance pathways, where overexpression confers resistance to oxaliplatin in colorectal cancer and 5-FU in breast cancer (Bhowmick et al., 2023; Chen et al., 2015; Huang et al., 2022). Prognostically, elevated baseline PAK6 correlated with shorter progression-free survival (PFS), independent of traditional STMs (Fig. 5A). Although prior studies found no PAK6-chemosensitivity association in non-SCLC models (Lei et al., 2022), our SCLC-specific data suggest tissue-context-dependent roles. Longitudinal PAK6 monitoring may thus aid in identifying high-risk patients and personalizing therapeutic regimens.

Without a doubt, this study has several limitations: (1) Single-center design and modest sample size constrain generalizability; (2) Lack of immunohistochemical validation limits tissue-level insights; (3) Insufficient overall survival (OS) data necessitate prospective follow-up. Our future research will focus on the following key directions: (1) Lentiviral modulation of PAK6 in SCLC models to define its roles in proliferation, invasion, and chemoresistance; (2) Preclinical evaluation of PAK6 inhibitors in patient-derived xenografts; (3) Multi-center trials assessing PAK6’s utility in chemoresistance prediction and subtype stratification (e.g., cisplatin-resistant vs. sensitive SCLC).

Conclusions

We identify PAK6 as a multi-faceted biomarker for SCLC with diagnostic, prognostic, and therapeutic monitoring value. Its cost-effective ELISA quantification facilitates clinical translation. While phased implementation in high-risk subgroups is recommended, broader adoption requires validation through prospective trials. Integrating PAK6 with emerging technologies (e.g., machine learning-assisted diagnostics: Chen et al., 2023; Wang et al., 2024) could further refine SCLC management paradigms.

Supplemental Information

Supplemental Information 1 The expression levels of STMs in LS-SCLC and ES-SCLC

Expression levels of PAK6(A), NSE (B), ProGRP(C), CEA (D), and CA19-9 (E) in LS-SCLC and ES-SCLC groups,There was no statistical difference between all marker groups.

Supplemental Information 2 The correlation between the serum levels of STMs at the initial diagnosis of SCLC and the post-treatment status of SCLC

The correlation between the serum levels of PAK6(A), NSE (B), ProGRP(C), CEA (D), and CA19-9 (E) at the initial diagnosis of SCLC and the post-treatment status of SCLC, categorized as partial response/stable disease (PR/SD) or progressive disease (PD). the level of PAK6 was significantly associated with treatment response Error bars represent median and interquartile range. (∗∗P ¡ 0.01,∗∗∗P ¡ 0.001)

Supplemental Information 3 Supplementary tables

Supplemental Information 4 Experimental analysis of the original data

Supplemental Information 5 The original results and conversion for PAK6 of ELISA experiment

Additional Information and Declarations

Competing Interests

Author Contributions

Human Ethics

Ethics

Data Availability

The authors declare there are no competing interests.

Simei Chen conceived and designed the experiments, performed the experiments, analyzed the data, authored or reviewed drafts of the article, and approved the final draft.

Kexin Han performed the experiments, prepared figures and/or tables, and approved the final draft.

Yinyi Chen analyzed the data, prepared figures and/or tables, and approved the final draft.

Liping Wei analyzed the data, prepared figures and/or tables, and approved the final draft.

Xinlu Sun performed the experiments, prepared figures and/or tables, and approved the final draft.

Yi Luo performed the experiments, prepared figures and/or tables, and approved the final draft.

Lili Wen performed the experiments, prepared figures and/or tables, and approved the final draft.

Liming Tan conceived and designed the experiments, authored or reviewed drafts of the article, and approved the final draft.

The following information was supplied relating to ethical approvals (i.e., approving body and any reference numbers):

The Second Affiliated Hospital of Nanchang University granted Ethical approval.

The following information was supplied relating to ethical approvals (i.e., approving body and any reference numbers):

Ethics Committee of the Second Affiliated Hospital of Nanchang University approval the study.

The following information was supplied regarding data availability:

The data are available in the Supplemental Files.

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
