# Peer review of "P21 activated kinase 6: a promising tool for predicting small cell lung cancer diagnosis and treatment response"

_PeerJ, doi:10.7717/peerj.19714_

## Round 0.1 · original submission · Minor Revisions

Per PeerJ policy, data must be provided in a machine-readable format (I suggest Zenodo: https://peerj.com/about/policies-and-procedures/cs). I recommend making the Figure 6 panels larger for readability.

Reviewer 1 ·

Basic reporting

1.First of all, logically speaking, the order of treatment response and diagnosis should be reversed in the title.

2.In Introduction section,the authors mention that PAK6 is highly expressed in many tumor groups and is associated with the development of many tumors. The role of PAK6 in SCLC is not mentioned, and the sudden introduction of the topic of expression of PAK6 in serum of SCLC patients has insufficient evidence support, which is slightly abrupt.

3. line 269 There is an extra “( )” in line 269.

Experimental design

1.(line 100-101)This cohort included 101 patients with SCLC and 92 patients with non-small cell lung cancer (NSCLC). But in most cases, NSCLC accounts for a high proportion of lung cancer, The inclusion and exclusion criteria need more detail.

2. Among the SCLC group, paired archived pre-treatment and post-treatment serum samples were collected,the data from other groups such as the NSCLC pre-treatment and post-treatment serum samples were not collected and analyzed.

Validity of the findings

1. From line 207-208, We know that serum levels of PAK6, ProGRP, and NSE significantly decreased following SCLC treatment.But the curative effect was divided into PR, SD and PD groups. Whether PAK6 level decreases in all groups after treatment?If it is so, it contradicts the conclusion that PAK6 can predict curative effect.Please explain it.

2.(line 210-212)The levels of PAK6 in the PR, SD, and PD groups were 51.99 (45.47-56.52) ng/L, 60.09 (52.72-69.52) ng/L, and 62.11 (54.10-69.15) ng/L, respectively. The PAK6 levels in the PR group were significantly lower than those in the SD and PD groups (p < 0.05, (Figure 5A). I think the differences between serum values cannot fully represent inter-group differences.We know that the total sample size is 101, but how many people are in the PR, SD and PD groups respectively.If too few samples are in a certain group ,will it affect the statistical results? If we change the unit (ng/L), will the inter-group differences become smaller?

Additional comments

1.In Figure1,we found that NSE and ProGRP differentially expressed in PN and NC .Can you explain it in discussion.

Reviewer 2 ·

Basic reporting

No comment

Experimental design

Progression free survival (PFS) is usually determined through clinical data and imaging. The authors have failed to define this according to normal standards. This leaves an impression that the PFS was not determined in the right manner to advocate the prognostic value of PAK6.

The abstract lacks a brief introductory statement to why the authors are looking at PAK6, ie what is the main research question to be answered?

Validity of the findings

While the findings are impressive, the authors have failed in their discussion to discuss what is the accessibility to this test in the public and how is this finding going to translate / inform clinical practice? Do the authors advocate testing these in all patients with SCLC?

The discussion section is too bulky and difficult to read. Each paragraph should only contain ONE main finding from their research and support / contrast the point with available literature / evidences. I would suggest to break down the paragraphs so that the reader can follow the author's train of thoughts.

---

## Round 0.2 · accepted · Accept

Thank you for addressing the reviewers' comments. We now believe this manuscript is ready for publication.

Reviewer 1 ·

Basic reporting

The article is clear and unambiguous.

Experimental design

The experimental design is reasonable.

Validity of the findings

This article has certain guiding significance for the article.

Additional comments

no comment

Reviewer 2 ·

Basic reporting

No comment

Experimental design

No comment

Validity of the findings

No comment